# Enhancement of Bone Ingrowth into a Porous Titanium Structure to Improve Osseointegration of Dental Implants: A Pilot Study in the Canine Model

**DOI:** 10.3390/ma13143061

**Published:** 2020-07-08

**Authors:** Ji-Youn Hong, Seok-Yeong Ko, Wonsik Lee, Yun-Young Chang, Su-Hwan Kim, Jeong-Ho Yun

**Affiliations:** 1Department of Periodontology, Periodontal-Implant Clinical Research Institute, School of Dentistry, Kyung Hee University, 26, Kyungheedae-ro, Dongdaemun-gu, Seoul 02447, Korea; jkama7@gmail.com; 2Department of Periodontology, College of Dentistry and Institute of Oral Bioscience, Jeonbuk National University, 567, Baekje-daero, Deokjin-gu, Jeonju-si, Jeollabuk-do 54896, Korea; dentquartz@naver.com; 3Advanced Process and Materials R&D Group, Korea Institute of Industrial Technology, 7-47 Songdo-dong, Yeonsu-gu, Incheon 406-840, Korea; wonslee@kitech.re.kr; 4Department of Dentistry, Inha International Medical Center, 424, Gonghang-ro, 84-gil, Unseo-dong, Jung-gu, Incheon 22382, Korea; bewitme@naver.com; 5Department of Periodontics, Asan Medical Center, 88, Olympic-ro 43-gil, Songpa-gu, Seoul 05505, Korea; suhwank@gmail.com; 6Department of Dentistry, University of Ulsan College of Medicine, 88, Olympic-ro 43-gil, Songpa-gu, Seoul 05505, Korea; 7Research Institute of Clinical Medicine of Jeonbuk National University-Biomedical Research Institute of Jeonbuk National University Hospital, 20, Geonjiro, Deokjin-gu, Jeonju-si, Jeollabuk-do 54907, Korea

**Keywords:** porosity, dental implant, osseointegration, bone formation, titanium

## Abstract

A porous titanium structure was suggested to improve implant stability in the early healing period or in poor bone quality. This study investigated the effect of a porous structure on the osseointegration of dental implants. A total of 28 implants (14 implants in each group) were placed in the posterior mandibles of four beagle dogs at 3 months after extraction. The control group included machined surface implants with an external implant–abutment connection, whereas test group implants had a porous titanium structure added to the apical portion. Resonance frequency analysis (RFA); removal torque values (RTV); and surface topographic and histometric parameters including bone-to-implant contact length and ratio, inter-thread bone area and ratio in total, and the coronal and apical parts of the implants were measured after 4 weeks of healing. RTV showed a significant difference between the groups after 4 weeks of healing (*p* = 0.032), whereas no difference was observed in RFA. In the test group, surface topography showed bone tissue integrated into the porous structures. In the apical part of the test group, all the histometric parameters exhibited significant increases compared to the control group. Within the limitations of this study, enhanced bone growth into the porous structure was achieved, which consequently improved osseointegration of the implant.

## 1. Introduction

A dental implant has been accepted as a reliable treatment modality for edentulous ridge with high long-term survival [1], and improvements in implant design, surface treatment, and surgical technique led to a marked increase in implant stability [2,3]. However, the results are mostly based on the selection of subjects with the exclusion of any clinical conditions that might have a negative effect on the healing around the implants. There are several possible risk factors associated with early implant failure or impaired healing, including smoking, head and neck radiation [4,5], bone quality and osteoporosis [6]. Osseointegration was defined as a direct and functional connection between bone and an artificial implant. However, the macroscopic (body structure and thread geometry) and microscopic (chemical composition and surface treatment) characteristics of dental implants could influence the success of the osseointegration [7].

The topographical features in an implant surface can be defined in terms of their scales, which were produced by surface modification treatments such as titanium plasma-spraying, grit-blasting, acid-etching, or combinations [2,8]. Apart from the macro-level that is related to the threaded screw of implant geometry and macroporous surface, mechanical interlocking is maximized by microtopographic roughness [9]. In addition, nanotopography is associated with the biological activities of cells to stimulate bone formation on an implant surface [10]. However, the majority of current surface treatments are unreliable to achieve reproducible nanoscale features as they range randomly from nanometers to millimeters.

Another approach in surface modification was the production of porous bodies of titanium metal and its alloys, and sintering of metal powders onto the surface was commonly used for porous coatings [11,12]. Advantages of porous surface-enhanced implant include the induction of new bone tissue ingrowth and neovascularization into the porous scaffold in three-dimensional (3D) aspects [13], elastic modulus closer to the cancellous bone that allows load distribution [14], and enhanced transport of metabolites and space for new bone through substantial porous interconnectivity [15]. To maximize the potential benefits from the porous structures, precise control of overall porosity and pore size was identified as important, although the optimal ranges were yet to be determined [16]. However, conventional methods had limitations of low volumetric porosity, irregular dimensions of pores, and poor interconnectivity [17]. Furthermore, possible mechanical failures related to a lack of yield strength and separation of coating materials led to soft tissue encapsulation and loosening of implants [18].

Recent approaches have utilized methods such as selective melting with laser or electron beam, 3D printing, casting or vapor deposition to control the internal pore geometry and distribution [12,19]. The porous scaffolds were sometimes combined with threaded implants for additional advantages in terms of primary mechanical stability and removability. One of the products that had been widely studied to adapt the porous structure to the root form implant was the porous tantalum trabecular metal (PTTM) enhanced implant [18,20,21]. PTTM was fabricated by foam-like vitreous carbon scaffold that resulted in the open-cell structure similar to the trabecular bone [18]. The PTTM part was added to the middle portion of the implant by laser welding and was combined with the screw-type design of titanium alloy surface at the cervical and apical portions, which were microtextured by grit-blasting with hydroxyapatite particles. From the animal studies, histomorphometric evaluations have demonstrated more new bone growth at the PTTM occupying the middle portion compared to the conventional surface within the 12-week study periods, and suggested the potential benefits of the porous structures in the compromised bone quality. However, there were limited biomechanical improvements assessed by the resonance frequency analysis in PTTM and the implant stability appeared comparable to the conventional microtextured surface.

In the present study, a novel method of utilizing the powder injection molding technique has been employed to form a porous titanium structure, which was fabricated on the apical portion of the machined screw-type implant. The effect of the newly developed porous structure on osseointegration was compared to the smooth-surfaced implant in the canine model.

## 2. Materials and Methods 

### 2.1. Design of the Implants

A threaded machined surface implant (c.p. titanium grade 4) with an external-type abutment connection (MegaGen Implant Co., Ltd., Gyeongsan, Korea) was used in the control group. The implant measured 4.1 mm in diameter, 8.1 mm in length, and had a straight configuration of the implant body (core diameter of 3.25 mm) with a homogenous thread height of 0.35 mm (Figure 1a). The test group implant had a porous titanium structure fabricated on the implant core at the apical portion; the core had a reduced diameter of 1.25 mm and a thread height of 1.35 mm to afford the space for the porous scaffold (1 mm in depth and 0.83 mm in width) (Figure 1b). The resulting profile had 3-mm spiral shape threads in the apical part, a regular pitch distance of 1.25 mm and a thread angle of 45°.

### 2.2. Fabrication of Porous Titanium Structure 

The porous titanium structure on the implant core was fabricated at the Korea Institute of Industrial Technology (KITECH) using insert powder injection molding technology (Figure 2a). Briefly, a feedstock, which was a mixture of titanium hydride (TiH_2_) powder, space holder and some polymeric binders, was prepared as a material for powder injection molding. Expandable polystyrene (EPS) beads with an average diameter of 325 µm were selected as space holders to form open-pore structure made from the contact between the beads during the expansion that occurred above 80 °C. The feedstock was injected into the narrow cavity between the threads at the apical third portion of the implant insert. The molded implants were inserted again into a mold designed for expansion of the EPS beads and kept for 20 min in an oven at 110 °C. The expanded beads were removed in a solvent, resulting in an open-pore structure consisting of TiH_2_ powder and binders (Figure 2b). The polymeric binders were removed completely during the thermal debinding process, slowly increasing the temperature up to 700 °C under argon atmosphere. During this process, TiH_2_ powder was also transformed to Ti powder by dehydrogenation reaction that occurred in the temperature range of from 350 to 500 °C. Finally, the open-pore scaffold of titanium powder between the threads was sintered for 3 h at 1100 °C in high vacuum (Figure 2c). During sintering, the titanium scaffold and implant core were combined into a single body by interdiffusion of titanium atoms at the interface, and titanium fixtures with open-pore titanium structure between the implant threads were formed. The porous structure had an average porosity of 68.1%, an average strut thickness of 61.4 µm (range: 30 to 100 µm), and an average pore size of 243 µm (range: 200 to 350 µm). The interconnected area between the pores that acts as the path for ingrowth of bone was measured to be 122.0 µm (range: 50 to 235 µm) in mean diameter from 2-dimensional image analysis. 

### 2.3. Animal Experiment

Four 12-month-old male beagle dogs weighing 12.0 to 17.0 kg were used in this study. The dogs were kept in separate cages under standard laboratory conditions. Animal selection, management, surgical procedures, and preparations were performed according to the protocols approved by the Institutional Animal Care and Use Committee at Korea Animal Medical Science Institute, Guri, Korea (Approval No. 16-KE-234). The study was conducted following the Animal Research: Reporting In Vivo Experiments (ARRIVE) guidelines [22].

Surgical procedures were performed under general anesthesia with intravenous injection of a solution (0.1 mL/kg) containing 1:1 ratio of tiletamine/zolazepam (Zoletil 50, Virbac S.A., Virbac Laboratories 06516, Carros, France) and xylazine hydrochloride (Rumpun, Bayer, Seoul, Korea). Infiltration anesthesia with 2% lidocaine HCl with 1:100,000 epinephrine (Huons, Seoul, Korea) was used at the surgical sites. The premolars (P1–P4) and the first molar (M1) in both the mandibles were carefully extracted and a total of 28 implants (14 implants for each group) were placed after 12 weeks. In each quadrant, three or four implants from the control or test group were randomly allocated. After sequential osteotomies, implants were installed under 40 Ncm (newton centimeter) of torque and submerged (Figure 3a–c). Antibiotics and nonsteroidal anti-inflammatory drugs were administered for 5 days. Sutures were removed after one week and animals were sacrificed after 4 weeks of healing by intravenous injection of 1 mL of suxamethonium chloride (50 mg/mL).

### 2.4. Resonance Frequency Analysis (RFA), Removal Torque Test and Topographical Analysis

Implant stability quotient (ISQ) value was measured immediately after the implant placement and at the time of animal sacrifice. A SmartPeg (Type 04, REF 100350, Osstell AB, Gothenburg, Sweden) was connected to each implant and a commercially available RFA equipment (Osstell Mentor, Osstell AB, Gothenburg, Sweden) was adjusted at the mesial and buccal direction of the implant (Figure 3a). The mean ISQ values in the mesial and buccal direction were recorded.

In each group, 7 implants were randomly selected and removal torque values (RTV) were measured using torque meter (MARK-10 torque gauge, MARK-10 Corporation, NY, USA) on the day of sacrifice. Removed implant specimens were dehydrated in graded ethanol series and sputter-coated with platinum (LEICA EM ACE200, sputter current 40 mA, Leica Microsystems, Wetzlar, Germany). Surface topography was examined under a field-emission scanning electron microscope (FE-SEM, SUPRA 40VP, Carl Zeiss, Oberkochen, Germany) and photograph images were taken at the magnifications of 20× and 100× with 5.0 kV.

### 2.5. Histologic and Histometric Analysis

Among the 14 implants allocated for each group, 7 specimens were processed for histologic and histometric analysis, as the rest of the 7 implants were tested for RTVs described in Section 2.4. Specimens of the implants and surrounding tissues were dissected into blocks and fixed in 10% buffered formalin solution. After sequential ethanol dehydration, nondecalcified specimens were embedded in methylmethacrylate (Technovit 7200, Kulzer GmbH, Hanau, Germany) and sectioned along the implant axis in the bucco-lingual plane using a diamond saw with 30–50 μm thickness. Hematoxylin and eosin (H&E)-stained sections were evaluated under a light microscope fitted with a camera and histometric measurements were completed using an automated image analysis program (Image-Pro Plus, Media Cybernetics, Rockville, MD, USA).

Parameters were measured from two parts (apical and coronal) of the implant, which was transversely divided along its long axis. The apical part included the area between the most apical border of the fixture and 3 mm above and coronal part was from the coronal border of the apical part to the most coronal endpoint thread of the fixture (Figure 4a). The following parameters were measured: (a) the bone-to-implant contact length (BICL, in mm), which was the sum of length of the implant surface in direct contact with surrounding bone; (b) the bone-to-implant contact ratio (BICR, in %), which was the percentage of BICL out of the length measured for the implant surface outline; (c) the inter-thread bone area (BA, in mm^2^), which was the sum of bone area observed between the threads; and (d) the inter-thread bone area ratio (BAR, in %), which was the percentage of BA in the region of interest (ROI). ROI in the control group and coronal part of the test group was determined by outlining the space between the threads (inter-thread space). To determine the corresponding ROI in the apical part of the test group, superimposition of the counterpart in the control implant was performed to outline the virtual boundary of the original shape of the inter-thread area (Figure 4b).

### 2.6. Statistical Analysis 

Statistical analyses were performed using SPSS Ver. 12.0 (SPSS, Chicago, IL, USA). Normality of data distribution was determined by Shapiro–Wilk test. For ISQ and RTV in both groups, paired t-test was used to compare the differences of the parameters between the two groups at each time period and the differences in ISQ between the baseline and 4 weeks in each group. Regarding histometric parameters, comparisons between the groups in each part and between the two parts (coronal and apical parts) in each group were performed using Student’s t-test (in the data of BICL and BICR in the apical part, BA and BAR in the coronal, apical and total area), or Wilcoxon’s rank-sum test (in the data of BICL and BICR in apical part and total area). The level of statistical significance was set at *p* < 0.05.

## 3. Results

### 3.1. Clinical Findings

The experimental sites in all the animals demonstrated uneventful healing and did not exhibit any adverse reaction throughout the postoperative healing period.

### 3.2. Resonance Frequency Analysis and Removal Torque Value 

No significant differences were observed in ISQ values between the two groups at each time period and between the baseline and 4 weeks in each group. However, RTV in the test group (20.5 ± 6.8) was higher than that of the control group (8.0 ± 3.6) with a significant difference at the 4-week healing period (*p* = 0.03) (Table 1).

### 3.3. Surface Topography from FE-SEM Images 

The original surface topography of the test group implant showed a titanium structure with regular distribution of pores in similar size ranging from 200 to 350 µm in the apical part (Figure 5a,b), whereas the control group showed a smooth texture of the machined surface (Figure 5c,d). After the removal torque test, the porous body in the test group exhibited structural destruction with a few integrated bone tissues in close proximity (Figure 5e,f). However, the interface between the porous structure and core of the fixture was maintained. The control group implant did not show any specific destruction, although there were some traces of scrapes on the surface (Figure 5g,h).

### 3.4. Histologic Analysis

New bone (NB) formation and BIC were observed at entire length of the implants in both the test and control groups (Figure 6a,f). In the coronal part of the test group, newly formed hard tissues projected from the parent bone (PB) surface towards the drilled osteotomy sites along the threads of the implant (Figure 6b,c). Osteoid and NB lined with osteoblast-like cells were found in the space between the threads, which were in direct contact with the implant surface and sometimes bridged PB with the implant surface. Some part of the PB surface underwent a remodeling process with reversal lines parallel to the long axis of the bone wall. These histologic features were also observed at both the coronal (Figure 6g,h) and apical parts (Figure 6i,j) of the control group which had the same surface topography of the coronal part in the test group. In the apical part of the test group, newly formed woven bone with osteoblast-like cells on its surface projected from the PB wall into the drilled space, and ingrowth of NB into the porous structure exhibited direct contact with the implant surface (Figure 6d). NB shown in the porous structure was integrated with the pore entrances and in direct contact with the inner surfaces of regularly distributed porous scaffolds (Figure 6e). NB was bridged between one another or to the PB surface and surrounded by the densely packed connective tissue matrix.

### 3.5. Histometric Analysis

Significant increases in BICL (*p* < 0.007), BICR (*p* = 0.011), BA (*p* = 0.014) and BAR (*p* = 0.028) of the total area were shown in the test group compared to the control group. The apical part of the test group presented significant increases in BICL (*p* < 0.001), BICR (*p* = 0.001), BA (*p* = 0.011) and BAR (*p* = 0.020) compared to the control group; all the parameters in the coronal part were similar in both the groups. The test group also showed significant differences between the coronal and apical part in BICL (*p* = 0.005), BICR (*p* = 0.010), BA (*p* = 0.009), and BAR (*p* = 0.049), whereas the control group showed no differences between the two parts (Table 2).

## 4. Discussion

In the present study, the porous titanium structure fabricated at the apical portion of the implant resulted in the interconnected open-pore structure, which led to an increase in the implant surface and its osteoconductivity to enhance bone ingrowth into the porous scaffold, thereby improving osseointegration of the implant.

The percentage of porosity on the overall surface and the size of pores are known as the determining factors in bone ingrowth [23]. Conventional methods like the sintering of beads on the titanium alloys have been reported to have limited degree of porosity (around 35%) and exhibit difficulty in controlling the profile of the topography [11]. As per the recent approaches, the PTTM- enhanced titanium implant could exhibit an increased percentage of the porosity (up to 70–80%) owing to the open-cell structure of dodecahedral repeats resembling trabecular bone [17,24]. In animal studies, the porous tantalum implants showed greater bone-to-implant contact with increased osteogenic activity compared to the solid titanium [20,25]. The clinical benefits of porous tantalum implants were also reported in the retrospective studies as there were high survival rates and less peri-implant bone loss [26], and a pilot study of failed implants immediately replaced by using the porous tantalum implants showed successful outcomes in 5 years of follow-up as well although the sample size was limited [27]. However, there still existed difficulty in manipulation of pure tantalum and high costs for purification [28]. 

The powder injection molding technique was introduced to process the fine ceramics in the past two decades and could offer the reproducible mass production of complicated structures like near-net-shapes, even in hard materials like ceramics [29]. In dental fields, powder injection molding zirconia implants were tested in the animal model, and this technique was suggested to provide an enhanced tissue response to the roughened surface when compared with that of a machined titanium surface [30]. In the present study, porous titanium structure produced by the powder injection molding technique resulted in the formation of interconnected open pores with an average porosity of about 70%. The structure was similar to the alveolar bone of the human mandible with D1 type bone (showing primarily of homogenous dense cortical bone) as evaluated through micro-computed tomography study, thus providing a negative template with a natural trabecular pattern with higher bone density [31]. Potential benefits of rapid vascularization, osteoblastic differentiation, and new bone ingrowth could be expected, as they mimic the natural trabecular bone. As the porosity could be controlled by adjusting the amount of polymers or the temperature of EPS expansion, it might also be possible to alter the characteristics of porous structures to match with the natural bone, or customized as per the need of individual patients. 

The multithreaded root-form implant has clinical benefits of simple osteotomy, implant placement, close mechanical proximity to the bone to increase primary stability, and less traumatic retrieval under conditions of failure [3]. When combined with the adequately controlled porous structure, additional effects of enhanced neovascularization and new bone formation inside the porous scaffold termed as osseoincoporation can be expected [32,33]. However, the porous structure in the present study has possible problems, such as a higher risk of bacterial plaque accumulation, mucosal and peri-implant diseases when compared to the machined surface upon exposure to complex oral environments [34]. Hybrid surface implants have been suggested to reduce the prevalence of peri-implantitis by including the machined surface or less roughened texture in the coronal part of the implant together with the rough surface treatments in the apical part, which played important role in healing between the bone and the surface [35]. In correspondence with the rationale of the hybrid implant, the test group implant was expected to have both advantages in terms of accelerated healing and increased bone-to-implant contact by the porous structure at the apical part, and less biological complications related to the inflammation at the coronal smooth surface area. In addition, an external connection was utilized to minimize the risk of fracture from the thinned lateral wall of the body after the decrease in core diameter, while providing space for the porous structure. The crestal bone–implant interface is an important area in stress distribution during load transmission [36], and there exists a lack of data regarding mechanical failure in the porous structure with reference to stress and fatigue when loading, thus stating inappropriateness of this structure in the coronal portion of the implant body. 

Implant stability measured using RFA showed ISQ values over 60 in both the groups and at both the observation periods with no significant differences. The ISQ value over 60 was reported to demonstrate clinical stability of the implant despite the differences in implant designs, surgical models, and devices used, and the factors affecting RFA include stiffness of the implant-bone interface, distance to first bone contact, and marginal bone loss [37]. It can be assumed that relatively standardized bone density among the surgical sites, fully engaged implant surface within the surrounding bone, and the same topographic aspects at the coronal portion contribute to the similar outcome of ISQ at baseline and after 4 weeks of healing between the two groups. The presence of a porous titanium structure in the apical portion and its loss of close proximity to the bone wall cut by the twist drill may not have a great influence on the ISQ value. On the other hand, the removal torque test is an indirect method to measure the shear strength that ruptures the bone–implant interface and gives information about the bone growth aspects at the surface [38]. Hence, a significant increase in RTV of the test group implant at 4 weeks might be explained by the increased secondary stability at the porous structure at the apical portion. The fractured bone compartment together with the porous surface in FE-SEM images after the reverse torque rotation also supported the findings of bone ingrowth at the apical portion. However, some of the destructions were observed in the center portion of the porous body, whereas the interface at the attachment to the implant core still remained. The destroyed sites might suggest the weak points in the mechanical properties of the implants, which should further be improved by the development of the manufacturing techniques. 

Significant increases in total BICL, BICR, BA, and BAR in the test group implants were attributed to the apical portion, and histologic findings supported the results by showing enhanced NB formation in direct contact with the interconnected open pores with an increased surface area at the apical portion and improved BIC at 4 weeks. Healing in the trabecular compartment relies on the process of osteoconduction and de novo bone formation at the implant surface, resulting in contact osteogenesis [33], and porous configuration could promote osteoconductivity by increasing space for blood clot stabilization and recruitment of various cells involved in biologic cascades of peri-implant healing [39]. A porous structure designed to resemble the trabecular patterns of highest bone quality might permit an adequate osteoconductive scaffold for bone ingrowth, which is in accordance with other studies [40,41]. Apparently, it also might help in providing enhanced micro-mechanical interdigitation with the parent bone and increase primary stability irrespective of the condition of the recipient site, including compromised bone quality such as D4 type bone (showing the fine trabecular bone composing almost the total volume) where the cortical stiffness could not be anticipated and the posterior maxilla that lacks bone height.

In the present study, a porous titanium structure fabricated by the powder injection molding technique was able to provide three-dimensional interconnected porosity on the implant surface and thereby enhanced the new bone ingrowth at the surface. The histometric findings, including the fact that the bone-to-implant contact and new bone formation inside the porous material demonstrated the improvements in osseointegration by the porous structure in the early healing dynamics, were in accordance with some other studies utilizing trabecular-like scaffolds to the dental implants [20,21,25]. However, the test implant used in this study was designed to combine the porous structure with the machined surface implant for a pilot approach to focus on the efficacy of the newly developed structure at the apical portion, and other variables, including overall macrogeometry and topography of the implants, were intended to be controlled. For the clinical applications, various modifications in the microstructures other than the machined surface at the coronal aspect should be considered, and it might also be necessary to develop the macrostructural designs of the implant–abutment connection and the platform that can show more favorable outcomes in the biomechanical aspects. Improvements in the fabrication technologies to standardize porosity and increase the mechanical strength of the porous structures are necessary to obtain reliable clinical outcomes. In addition, the healing events around the porous implants in the long-term period and the tissue dynamics after the loading should further be observed. Finally, the effects of the porous titanium structure on the compromised bone conditions in clinical situations such as osteoporosis, grafted bone or simultaneous sinus floor elevation should be further investigated.

## 5. Conclusions

The findings of the study suggest that the porous titanium structure might increase apical bone-to-implant contact due to the increased surface area and enhance new bone formation with increased osteoconductivity in the early healing period, thereby leading to improvements in the osseointegration of the implants.

## Figures and Tables

**Figure 1 materials-13-03061-f001:**
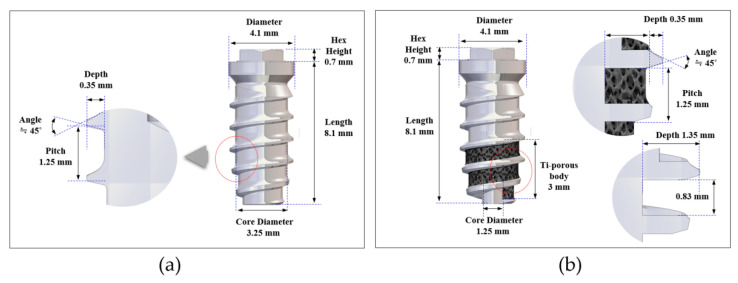
Design of the implants: (**a**) overall profiles and dimensions of the implant in the control group; (**b**) overall profiles and dimensions of the implant in the test group.

**Figure 2 materials-13-03061-f002:**
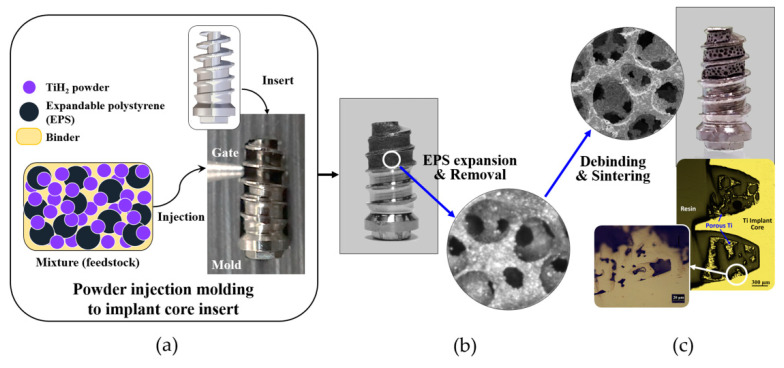
Insert powder injection molding process to fabricate the porous titanium fixture: (**a**) after a titanium implant with a narrow cavity in the deep inter-thread area was inserted into a mold, a mixture of titanium hydride (TiH_2_) powder, expandable polystyrene (EPS) beads and some polymeric binders were injected into the narrow cavity between the threads of the implant; (**b**) the EPS beads within the molded area were removed in a solvent after the expansion of EPS; (**c**) subsequently, through the debinding process, binders were removed and after dehydrogenation of TiH_2_ to Ti powder and the sintering process, the porous titanium fixtures with an open-pored titanium structure between the implant threads were produced and combined into a single body.

**Figure 3 materials-13-03061-f003:**
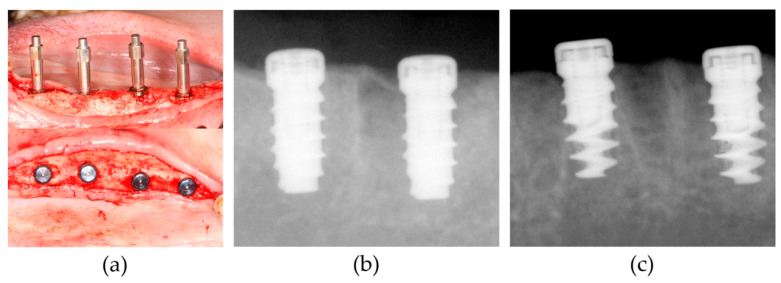
Surgical procedure of implant placement: (**a**) Smartpeg connection to measure implant stability quotient (ISQ) values using the resonance frequency analysis (RFA) device (Osstell Mentor) immediately after the implant placement (upper), and coverscrew connection (lower); (**b**) periapical X-ray images were taken in the control group; (**c**) periapical X-ray images were taken in the test group. Radiolucency near the implant surface was shown at the apical part of the test group compared to the control group.

**Figure 4 materials-13-03061-f004:**
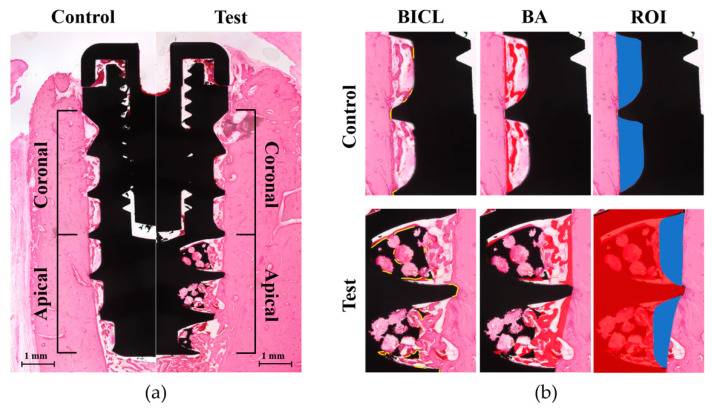
Schematic images of the histometric measurements: (**a**) Parameters were measured within the coronal and apical part, which were transversely divided along the long axis of the implant in both the control (left) and test groups (right) (magnification of 20×). (**b**) In each part of the control or test group (magnification of 50×), the bone-to-implant contact length (BICL) was determined by the total length of the implant surface in contact with the surrounding bone (yellow outlines in BICL). The inter-thread bone area (BA) was measured by the total bone area between the threads (red-colored area in BA). The region of interest (ROI) of the control group was outlined for the inter-thread space (blue colored area in ROI at the upper line). The ROI of the apical part in the test group (blue-colored area in ROI at the lower line) was determined by the virtual boundary, which was made from the superimposed counterpart of the control implant (red-colored implant shape in ROI at the lower line). Consequently, the original shape of the inter-thread space was outlined and the BA ratio (BAR) was calculated from the percentage of BA within each ROI.

**Figure 5 materials-13-03061-f005:**
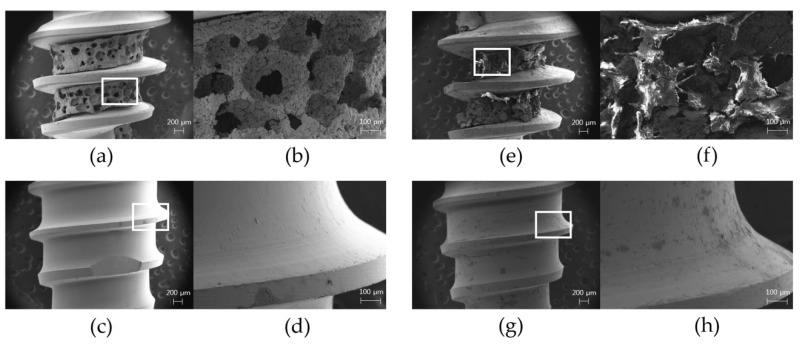
Field-emission scanning electron microscope (FE-SEM) images before the implant placement and after the removal: (**a**) overall image of the test group before the placement (magnification of 20×); (**b**) magnified view of the white box in (**a**) showing a porous titanium structure in the apical part of the test group implant with regularly distributed pores (magnification of 100×); (**c**) overall image of the control group before the placement (magnification of 20×); (**d**) magnified view of the white box in (**c**) showing a smooth machined surface in the control group implant (magnification of 100×); (**e**) overall image of the test group after the removal of implants at 4-week healing periods (magnification of 20×); (**f**) magnified view of the white box in (**e**) showing the destruction of the porous structures and some bone tissues at the porous structure after the removal (magnification of 100×); (**g**) overall image of the control group after the removal of implants at 4-week healing periods (magnification of 20×); (**h**) magnified view of the white box in (**g**) showing no specific destruction in the surface profile except for some scrapes after the removal (magnification of 100×).

**Figure 6 materials-13-03061-f006:**
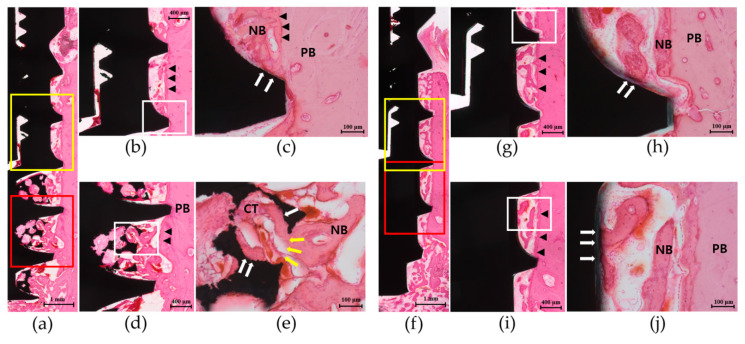
Representative histologic images of the implant in the test group and control group at 4 weeks of healing (H&E staining): (**a**) Overall view of the whole length of the test group implant showing a smooth surface profile in the coronal part (yellow box) and the porous structure in the apical part (red box) (magnification of 20×). (**b**) In the coronal part, new bone (NB) projection from the parent bone (PB) into the drilled osteotomy sites along the implant threads was observed (magnification of 50×). (**c**) Magnified view of the white box in (**b**) showing NB in direct contact with the implant surface (white arrows) (magnification of 200×). (**d**) In the apical part, NB connected with the PB surface and in contact with the surface of the porous structure was shown (magnification of 50×). (**e**) Magnified view of the white box in (**d**) showing NB ingrowth through the pore entrance (yellow arrows) and in direct contact with the surface of the porous scaffold (white arrows) (magnification of 200×). NB was lined with osteoblast-like cells on its surface and surrounded by the densely packed connective tissue matrix (CT). Reversal lines (black arrowheads) at the PB surface were found along the bony wall. (**f**) Overall view of the whole length of the control group implant showing a smooth surface profile in both the coronal (yellow box) and apical parts (red box) (magnification of 20×). (**g**) In the coronal part, new bone (NB) projected from the parent bone (PB) surface and towards the inter-thread space was shown (magnification of 50×). (**h**) Magnified view of the white box in (**g**) showing NB in direct contact with the implant surface (white arrows) and lined with osteoblast-like cells on its surface (magnification of 200×). (**i**) In the apical part, histologic appearance similar to that of the coronal part was shown (magnification of 50×). (**j**) Magnified view of the white box in (**i**) showing NB directly bridged to the implant surface (magnification of 200×). Lamellated reversal lines (black arrowheads) can be seen at the PB surface along the bony wall.

**Table 1 materials-13-03061-t001:** Implant stability quotient (ISQ) value and removal torque value (RTV) after 4 weeks of healing in the test and control groups (mean ± SD).

**ISQ Value**		**Test Group (n = 14)**	**Control Group (n = 14)**
	Baseline	66.7 ± 4.0	69.5 ± 8.3
	4 weeks	67.5 ± 5.0	68.4 ± 6.3
**RTV (Ncm)**		**Test Group (n = 7)**	**Control Group (n = 7)**
	4 weeks	20.5 ± 6.8 ^1^	8.0 ± 3.6

^1^ Statistically significant difference between the two groups in the paired t-test (*p* < 0.05).

**Table 2 materials-13-03061-t002:** Histometric parameters after 4 weeks of healing in the test and control groups (mean ± SD).

		Test Group (n = 7)	Control Group (n = 7)
**BICL (mm)**	Coronal part	1.88 ± 0.98	1.31 ± 0.76
	Apical part	3.43 ± 0.68 ^1,2^	1.30 ± 0.78
	Total	5.31 ± 1.28^1^	2.60 ± 1.21
**BICR (%)**	Coronal part	17.83 ± 9.53	12.56 ± 7.26
	Apical part	31.70 ± 7.53 ^1,2^	12.89 ± 7.57
	Total	24.83 ± 6.44^1^	12.69 ± 5.74
**BA (mm^2^)**	Coronal part	0.28 ± 0.11	0.21 ± 0.08
	Apical part	0.48 ± 0.12 ^1,2^	0.25 ± 0.12
	Total	0.77 ± 0.24 ^1^	0.45 ± 0.16
**BAR (%)**	Coronal part	24.9 ± 10.84	20.7 ± 7.74
	Apical part	37.2 ± 10.04 ^1,2^	23.0 ± 9.75
	Total	31.5 ± 8.03 ^1^	21.8 ± 6.36

BICL, the bone-to-implant contact length; BICR, the bone-to-implant contact ratio; BA, the inter-thread bone area; BAR, the inter-thread bone area ratio. ^1^ Statistically significant difference from the control group in Wilcoxon’s rank-sum test (used for BICL and BICR in total area) and in Student’s t-test (used for BICL, BICR, BA, and BAR in apical part, BA and BAR in total area) (*p* < 0.05). ^2^ Statistically significant difference from the coronal part in Student’s t-test (*p* < 0.05).

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
