# Peer review of "Enhancement of Bone Ingrowth into a Porous Titanium Structure to Improve Osseointegration of Dental Implants: A Pilot Study in the Canine Model"

_materials, 2020, doi:10.3390/ma13143061_

Round 1
Reviewer 1 Report
Experimental design is well presented and the results with figures support the statements. Figures 6 and 7 should be combined to compare the histology of both groups simultaneously. Higher power histology is too dark. Introduction into discussion should be redone since some irrelevant comments have not been removed.
Author Response
Dear Editor and reviewer (1)
We are very grateful to you and the journal’s reviewers for the critical comments and useful suggestions that have helped us to improve our paper considerably. We have taken all these feedbacks into account and submit a revised version of our manuscript.
Referee(s)' Comments to Author:
Reviewer 1
Experimental design is well presented and the results with figures support the statements.
Figures 6 and 7 should be combined to compare the histology of both groups simultaneously. Higher power histology is too dark.
Answers) Thank you very much for the comments. We have combined Figure 6 and 7 into one panel and also corrected the brightness of Figure 6c, 6e, 7c and 7e. New figure (Figure 6) was made as follows. Also, the figure legend of newly made Figure 6 was partly corrected.
Figure 6. Representative histologic images of the implant in the test group and control group at 4 weeks of healing (H & E staining): (a) Overall view of the whole length of the test group implant showing a smooth surface profile in the coronal part (yellow box) and porous structure in the apical part (red box) (magnification of 20×); (b) In the coronal part, new bone (NB) projection from the parent bone (PB) into the drilled osteotomy sites along the implant threads was observed (magnification of 50×); (c) Magnified view of the white box in (b) showing NB in direct contact with the implant surface (white arrows) (magnification of 200×); (d) In the apical part, NB connected with the PB surface and in contact with the surface of porous structure was shown (magnification of 50×); (e) Magnified view of the white box in (d) showing NB ingrowth through the pore entrance (yellow arrows) and in direct contact with the surface of the porous scaffold (white arrows) (magnification of 200×). NB was lined with osteoblast-like cells on its surface and surrounded by the densely packed connective tissue matrix (CT). Reversal lines (black arrowheads) at the PB surface were found along the bony wall; (f) Overall view of the whole length of the control group implant showing a smooth surface profile in both the coronal (yellow box) and apical parts (red box) (magnification of 20×); (g) In the coronal part, new bone (NB) projected from the parent bone (PB) surface and towards the inter-thread space was shown (magnification of 50×); (h) Magnified view of the white box in (g) showing NB in direct contact with the implant surface (white arrows) and lined with osteoblast-like cells on its surface (magnification of 200×); (i) In the apical part, histologic appearance similar to that of the coronal part was shown (magnification of 50×); (j) Magnified view of the white box in (i) showing NB directly bridged to the implant surface (magnification of 200×). Lamellated reversal lines (black arrowheads) can be seen at the PB surface along the bony wall.
Introduction into discussion should be redone since some irrelevant comments have not been removed.
Answers) Thank you very much for the comments and we are very sorry about the mistakes we have made. We have removed the sentence.
We hope that the revised version of our paper is now suitable for publication in Materials_Special Issue: Advances in Bone Graft Materials and we look forward to hearing from you at your earliest convenience.
Sincerely yours,
Jeong-Ho Yun, BS, DDS, MSD, PhD

Reviewer 2 Report
Reviewer’s Comments to the Authors
The purpose of this study was to develop a new screw-type implant with a porous titanium structure on its apical portion and to investigate the effect of the porous structure on the osseointegration of dental implants in a canine model. The authors describe the design and fabrication process of the implants and evaluate the quality of osseointegration compared to a smooth-surfaced implant using resonance frequency analysis, removal torque measurements, scanning electron microscopy, and histometric analysis.
Comments:
Introduction:
- Overall, the introduction is informative and to the point.
- The last sentence of the first paragraph (lines 52-53) implies that this study might actually compare osseointegration in a ‘normal’ patient vs. a patient with a compromised tissue condition, which is not the case. Therefore, this sentence should probably be rewritten or removed, especially since the evaluation of the porous titanium structure in a compromised bone condition model as a next step is later suggested by the authors at the end of the discussion (lines 394-396).
Materials and Methods:
- This section is well described. The only information that should be added to section ‘2.5. Histologic and histometric analysis’ is that 7 samples/implants were used for this analysis.
Results:
- The results are well structured, and all observations properly described.
- To facilitate the comparison of the histological images of the test and control group it would be better if Figures 6 and 7 could be combined into one panel.
- The white balance in Figures 6c, 6e, 7c, 7e needs to be corrected.
Discussion:
- Overall, the discussion is well written.
- The authors should remove the first paragraph of the discussion as it seems to be the description of what the Discussion should contain.
- Some of the references in this paper, and specifically in the Discussion appear to be quite old with only 3 references published within the last 5 years. It seems that there are many more recent studies published about implants with porous titanium structures and implants that were produced using insert powder injection technology that should be included.
- An implant with a hybrid surface is already on the market (https://www.en.ires.dental/implant-systems/imax/). What, according to the authors, are the advantages of their own design? Please add a comment to the discussion.
Author Response
Dear Editor and reviewer (2)
We are very grateful to you and the journal’s reviewers for the critical comments and useful suggestions that have helped us to improve our paper considerably. We have taken all these feedbacks into account and submit a revised version of our manuscript.
Referee(s)' Comments to Author:
Reviewer 2
The purpose of this study was to develop a new screw-type implant with a porous titanium structure on its apical portion and to investigate the effect of the porous structure on the osseointegration of dental implants in a canine model. The authors describe the design and fabrication process of the implants and evaluate the quality of osseointegration compared to a smooth-surfaced implant using resonance frequency analysis, removal torque measurements, scanning electron microscopy, and histometric analysis.
Comments:
Introduction:
Overall, the introduction is informative and to the point.
The last sentence of the first paragraph (lines 52-53) implies that this study might actually compare osseointegration in a ‘normal’ patient vs. a patient with a compromised tissue condition, which is not the case. Therefore, this sentence should probably be rewritten or removed, especially since the evaluation of the porous titanium structure in a compromised bone condition model as a next step is later suggested by the authors at the end of the discussion (lines 394-396).
Answer) Thank you very much for the comments. We do agree that the sentence mentioning about the osseointegration in the compromised tissue conditions might not be appropriate in the present step and in the introduction section. We have removed the sentence and the paragraph is as follows.
A dental implant has been accepted as a reliable treatment modality for edentulous ridge with high long-term survival [1], and improvements in implant design, surface treatment, and surgical technique led to a marked increase in implant stability [2,3]. However, the results are mostly based on the selection of subjects with the exclusion of any clinical conditions that might have a negative effect on the healing around the implants. There are several possible risk factors associated with early implant failure or impaired healing including smoking, head and neck radiation [4,5], bone quality and osteoporosis [6].
Materials and Methods:
This section is well described. The only information that should be added to section ‘2.5. Histologic and histometric analysis’ is that 7 samples/implants were used for this analysis.
Answer) Thank you very much for the comments. As a total of 14 implants were allocated in each group and 7 implants were tested for removal torque test (described in section 2.4.), the rest of the 7 implants were included in the histologic and histometric analysis. This was clarified with the addition of the following sentence as follows (in the section 2.5.).
Among the 14 implants allocated for each group, 7 specimens were processed for histologic and histometric analysis as the rest of the 7 implants were tested for RTVs described in section 2.4.
Results:
The results are well structured, and all observations properly described.
To facilitate the comparison of the histological images of the test and control group it would be better if Figures 6 and 7 could be combined into one panel. The white balance in Figures 6c, 6e, 7c, 7e needs to be corrected.
Answer) Thank you for the comments. We have combined Figure 6 and 7 into one panel and also corrected the brightness of Figure 6c, 6e, 7c and 7e. New figure (Figure 6) was made as follows. Also, the figure legend of newly made Figure 6 was partly corrected.
Figure 6. Representative histologic images of the implant in the test group and control group at 4 weeks of healing (H & E staining): (a) Overall view of the whole length of the test group implant showing a smooth surface profile in the coronal part (yellow box) and porous structure in the apical part (red box) (magnification of 20×); (b) In the coronal part, new bone (NB) projection from the parent bone (PB) into the drilled osteotomy sites along the implant threads was observed (magnification of 50×); (c) Magnified view of the white box in (b) showing NB in direct contact with the implant surface (white arrows) (magnification of 200×); (d) In the apical part, NB connected with the PB surface and in contact with the surface of porous structure was shown (magnification of 50×); (e) Magnified view of the white box in (d) showing NB ingrowth through the pore entrance (yellow arrows) and in direct contact with the surface of the porous scaffold (white arrows) (magnification of 200×). NB was lined with osteoblast-like cells on its surface and surrounded by the densely packed connective tissue matrix (CT). Reversal lines (black arrowheads) at the PB surface were found along the bony wall; (f) Overall view of the whole length of the control group implant showing a smooth surface profile in both the coronal (yellow box) and apical parts (red box) (magnification of 20×); (g) In the coronal part, new bone (NB) projected from the parent bone (PB) surface and towards the inter-thread space was shown (magnification of 50×); (h) Magnified view of the white box in (g) showing NB in direct contact with the implant surface (white arrows) and lined with osteoblast-like cells on its surface (magnification of 200×); (i) In the apical part, histologic appearance similar to that of the coronal part was shown (magnification of 50×); (j) Magnified view of the white box in (i) showing NB directly bridged to the implant surface (magnification of 200×). Lamellated reversal lines (black arrowheads) can be seen at the PB surface along the bony wall.
Discussion:
Overall, the discussion is well written.
The authors should remove the first paragraph of the discussion as it seems to be the description of what the Discussion should contain.
Answer) Thank you very much for the comments and we are very sorry about the mistakes we have made. We have removed the sentence.
Some of the references in this paper, and specifically in the Discussion appear to be quite old with only 3 references published within the last 5 years. It seems that there are many more recent studies published about implants with porous titanium structures and implants that were produced using insert powder injection technology that should be included.
Answer) Thank you very much for the comments. We have searched the recent papers about the porous implants and included some results about the clinical effects shown in the animal and human studies. As the porous tantalum trabecular metal (PTTM) enhanced titanium implants were one of the widely studied products in the recent dental fields, the updated references were mostly related to the PTTM implants. Although there were limited reports of powder injection molding techniques for dental implants, we have also added one of the study that was applied in animal model. These studies were mentioned in the discussion section as follows.
The percentage of porosity on the overall surface and the size of pores are known as the determining factors in bone ingrowth [20]. Conventional methods like sintering of beads on the titanium alloys has been reported to have limited degree of porosity (around 35%) and exhibit difficulty in controlling the profile of the topography [10]. As per the recent approaches, the porous tantalum trabecular metal (PTTM) enhanced titanium implant could exhibit an increased percentage of the porosity (up to 70-80%) owing to the open-cell structure of dodecahedral repeats resembling trabecular bone [16,21]. In animal studies, the porous tantalum implants showed greater bone-to-implant contact with increased osteogenic activity compared to the solid titanium [22,23]. The clinical benefits of porous tantalum implants were also reported in the retrospective studies as there were high survival rates and less peri-implant bone loss [24], and a pilot study of failed implants immediately replaced by using the porous tantalum implants showed successful outcomes in 5 year follow-ups as well although the sample size was limited [25]. However, there still existed difficulty in manipulation of pure tantalum and high costs for purification [26].
Powder injection molding technique was introduced to process the fine ceramics in the past two decades and could offer the reproducible mass production of complicated structures like near-net-shapes even in the hard materials as ceramics [27]. In dental fields, powder injection molding zirconia implants were tested in the animal model and suggested to have enhanced tissue response to the roughened surface fabricated with this technique compared with machined titanium surface [28]. In the present study, porous titanium structure produced by powder injection molding technique resulted in the formation of interconnected open pores with an average porosity of about 70%.
- new reference was added for the sentence :
- Lee, J.W.; Wen, H.B.; Gubbi, P.; Romanos, G.E. New bone formation and trabecular bone microarchitecture of highly porous tantalum compared to titanium implant threads: A pilot canine study. Clin Oral Implants Res 2018, 29, 164-174.
- Fraser, D.; Mendonca, G.; Sartori, E.; Funkenbusch, P.; Ercoli, C.; Meirelles, L. Bone response to porous tantalum implants in a gap-healing model. Clin Oral Implants Res 2019, 30, 156-168.
- Edelmann, A.R.; Patel, D.; Allen, R.K.; Gibson, C.J.; Best, A.M.; Bencharit, S. Retrospective analysis of porous tantalum trabecular metal-enhanced titanium dental implants. J Prosthet Dent 2019, 121, 404-410.
- Dimaira, M. Immediate placement of trabecular implants in sites of failed implants. Int J Oral Maxillofac Implants 2019, 34, e77-e83.
- Lin, S.I.E. Near-net-shape forming of zirconia optical sleeves by ceramics injection molding. Ceramics International 2001, 27, 205-214.
- Park, Y.S.; Chung, S.H.; Shon, W.J. Peri-implant bone formation and surface characteristics of rough surface zirconia implants manufactured by powder injection molding technique in rabbit tibiae. Clin Oral Implants Res 2013, 24, 586-591.
An implant with a hybrid surface is already on the market (https://www.en.ires.dental/implant-systems/imax/). What, according to the authors, are the advantages of their own design? Please add a comment to the discussion.
Answer) Thank you very much for the comments and suggestions. What we have expected with the present test group implant was to have advantages on the increased healing and osseointegration with porous structure at the apical portion, but also wanted to avoid or reduce biological complications related to the inflammation at the coronal part. When the porous structure was treated to the whole length of the implant, there might be higher risk of the prevalence of the plaque accumulation and peri-implantitis at the coronal portion and the benefits from the multithreaded implant also might be lost. Therefore, the hybrid design was planned to maximize the clinical benefits by combining these different structures. This was added in the third paragraph of the Discussion section as follows.
The multithreaded root-form implant has clinical benefits of simple osteotomy, implant placement, close mechanical proximity to bone to increase primary stability, and less traumatic retrieval under conditions of failure [3]. When combined with the adequately controlled porous structure, additional effects of enhanced neovascularization and new bone formation inside the porous scaffold termed as osseoincoporation can be expected [30,31]. However, the porous structure in the present study has the possible problems like higher risk of bacterial plaque accumulation, mucosal and peri-implant diseases compared to the machined surface upon exposure to complex oral environments [32]. Hybrid surface implants had been suggested to reduce the prevalence of peri-implantitis by including the machined surface or less roughened texture in the coronal part of the implant together with the rough surface treatments in the apical part which played important role in healing between bone and the surface [33]. In correspondence to the rationale of the hybrid implant, the test group implant was expected to have both advantages on the accelerated healing and increased bone-to-implant contact by porous structure at the apical part and less biological complications related to the inflammation at the coronal smooth surface area. In addition, external connection was utilized to minimize the risk of fracture from the thinned lateral wall of the body after the decrease in core diameter while providing space for the porous structure. Crestal bone-implant interface is an important area in stress distribution during load transmission [34], and there exists a lack of data about mechanical failure in the porous structure with reference to stress and fatigue when loading, thus stating inappropriateness of this structure in the coronal portion of the implant body.
- new reference was added for the sentence :
- Lee, C.T.; Tran, D.; Jeng, M.D.; Shen, Y.T. Survival rates of hybrid rough surface implants and their alveolar bone level alteration. J Periodontol 2018, 12, 1390-1399.
We hope that the revised version of our paper is now suitable for publication in Materials_Special Issue: Advances in Bone Graft Materials and we look forward to hearing from you at your earliest convenience.
Sincerely yours,
Jeong-Ho Yun, BS, DDS, MSD, PhD

Reviewer 3 Report
I hope that the results of this study will find clinical application as soon as possible.
Author Response
Reviewer 3: I hope that the results of this study will find clinical application as soon as possible.
Thank you very much for the kind comments. We are planning to have further investigations for the improvements in the design of the implant and more evidences that could support the clinical applications.

Reviewer 4 Report
The manuscript submitted to Materials entitled “Enhancement of bone ingrowth into porous titanium structure to improve osseointegration of dental implants: A pilot study in the canine model” is an original article which aim to investigate the effect of porous structure on the osseointegration of dental implants.
On my opinion the article is interesting, well written, with good English. The content of the manuscript is very interesting.
The authors suggested that the porous titanium structure might increase apical bone-to-implant contact resulting from the increased surface area and enhance new bone formation with increased osteoconductivity leading to improvements in osseointegration.
However, I highlighted some issues.
Introduction. Specify if there are other similar studies. Better specify the objectives and methods of the study.
Insert the following sentence at the end of line 52 on page 2:
<<Osseointegration has been defined as a direct and functional connection between bone and an artificial implant. However, the macroscopic (body structure and wire geometry) and microscopic (chemical composition and surface treatment) characteristics of dental implants could influence the success of these procedures [https://doi.org/10.23812/20-96-L-53].>>.
Discussion. Are there other similar studies that have shown similar results? Did the authors find limitations in their study by comparing it with other in the literature?
After making the indicated changes the article may be suitable for publication.
Author Response
Please see the attachment.
Reviewer 4
The manuscript submitted to Materials entitled “Enhancement of bone ingrowth into porous titanium structure to improve osseointegration of dental implants: A pilot study in the canine model” is an original article which aim to investigate the effect of porous structure on the osseointegration of dental implants. On my opinion the article is interesting, well written, with good English. The content of the manuscript is very interesting. The authors suggested that the porous titanium structure might increase apical bone-to-implant contact resulting from the increased surface area and enhance new bone formation with increased osteoconductivity leading to improvements in osseointegration. However, I highlighted some issues.
- Introduction. Specify if there are other similar studies. Better specify the objectives and methods of the study.
Answer> Thank you very much for the comments. There have been studies about the porous tantalum trabecular metal (PTTM) enhanced implant, which aimed to develop methods to utilize porous structures in the dental fields. The products were already been commercialized and the studies evaluated for the efficacy in new bone growth compared to the conventional microtextured surface were demonstrated. Therefore, we have added some relevant studies utilizing the porous structures in the dental implants as follows with new references (highlighted in light green).
Recent approaches have utilized methods such as selective melting with laser or electron beam, 3D printing, casting or vapor deposition to control the internal pore geometry and distribution [12,19]. The porous scaffolds were sometimes combined with threaded implants for additional advantages in terms of primary mechanical stability and removability. One of the products that had been widely studied to adapt the porous structure to the root form implant was the porous tantalum trabecular metal (PTTM) enhanced implant [18,20,21]. PTTM was fabricated by foam-like vitreous carbon scaffold that resulted in the open-cell structure similar to the trabecular bone [18]. The PTTM part was added to the middle portion of the implant by laser welding and was combined with the screw-type design of titanium alloy surface at the cervical and apical portion which were microtextured by grit-blasting with hydroxyapatite particles. From the animal studies, histomorphometric evaluations have demonstrated more new bone growth at the PTTM occupying the middle portion compared to the conventional surface within the 12-week study periods and suggested the potential benefits of the porous structures in the compromised bone quality. However, there were limited biomechanical improvements assessed by the resonance frequency analysis in PTTM and the implant stability appeared comparable to the conventional microtextured surface.
- Lee, J.W.; Wen, H.B.; Gubbi, P.; Romanos, G.E. New bone formation and trabecular bone microarchitecture of highly porous tantalum compared to titanium implant threads: A pilot canine study. Clin Oral Implants Res 2018, 29, 164-174.
- Fraser, D.; Funkenbusch, P.; Ercoli, C.; Meirelles, L. Biomechanical analysis of the osseointegration of porous tantalum implants. J Prosthet Dent 2020, 123, 811-820.
- Insert the following sentence at the end of line 52 on page 2:
<<Osseointegration has been defined as a direct and functional connection between bone and an artificial implant. However, the macroscopic (body structure and wire geometry) and microscopic (chemical composition and surface treatment) characteristics of dental implants could influence the success of these procedures [https://doi.org/10.23812/20-96-L-53].>>.
Answer> Thank you for your kind suggestion and the reference to help improve this paper. We have added the sentence and the reference below in the manuscript page 2 line 52-55 (highlighted in light green).
- Giudice, A.; Bennardo, F., Antonelli, A., Barone, S., Wagner, F., Fortunato, L., Traxler, H. Influence of clinician’s skill on primary implant stability with conventional and piezoelectric preparation techniques: an ex-vivo study. J Biol Regul Homeost Agents 2020, 34, 739-745.
- Discussion. Are there other similar studies that have shown similar results? Did the authors find limitations in their study by comparing it with other in the literature?
Answer> Thank you very much for the important comment and suggestion. There have been some previous other studies that have utilized the porous structures on the implant surface and have shown similar tendencies in the increased new bone ingrowth into the porous materials, and thereby improved the osseointegration in early healing periods. However, there are some differences in the design of the test implant as the present study was a pilot approach to evaluate the efficacy of porous structure under the control of other variables including the microtextures resulting from other surface treatments. The last paragraph of the discussion was rephrased and corrected as follows to describe the comparisons with other studies and limitations within the present study (highlighted in light green).
In the present study, porous titanium structure fabricated by the powder injection molding technique could provide three-dimensional interconnected porosity on the implant surface and thereby enhanced the new bone ingrowth at the surface. The histometric findings including the bone-to-implant contact and new bone formation inside the porous material have demonstrated the improvements of the osseointegration by the porous structure in the early healing dynamics, which were in accordance with some other studies utilizing the trabecular-like scaffolds to the dental implants [20,21,25]. However, the test implant used in this study was designed to combine the porous structure with the machined surface implant for a pilot approach to focus on the efficacy of the newly developed structure at the apical portion and other variables including overall macrogeometry and topography of the implants were intended to be controlled. For the clinical applications, various modifications in the microstructures other than the machined surface at the coronal aspect should be considered and it might be necessary to develop the macrostructural designs of the implant-abutment connection and the platform that can show more favorable outcomes in the biomechanical aspects as well. Improvements in the fabrication technologies to standardize porosity and increase the mechanical strength of the porous structures are necessary to obtain reliable clinical outcomes. In addition, the healing events around the porous implants in the long-term period and the tissue dynamics after the loading should further be observed. Finally, the effects of the porous titanium structure on the compromised bone conditions in clinical situations such as osteoporosis, grafted bone or simultaneous sinus floor elevation should be further investigated.
